# Facial Appearance and Psychosocial Features in Orthognathic Surgery: A FACE-Q- and 3D Facial Image-Based Comparative Study of Patient-, Clinician-, and Lay-Observer-Reported Outcomes

**DOI:** 10.3390/jcm8060909

**Published:** 2019-06-25

**Authors:** Rafael Denadai, Pang-Yun Chou, Yu-Ying Su, Chi-Chin Lo, Hsiu-Hsia Lin, Cheng-Ting Ho, Lun-Jou Lo

**Affiliations:** 1Department of Plastic and Reconstructive Surgery and Craniofacial Research Center, Chang Gung Memorial Hospital, Chang Gung University, Taoyuan 333, Taiwan; denadai.rafael@hotmail.com (R.D.); chou.asapulu@gmail.com (P.-Y.C.); samuelyaya98@gmail.com (C.-C.L.); 2Department of Craniofacial Orthodontics and Craniofacial Research Center, Chang Gung Memorial Hospital, Taoyuan 333, Taiwan; suyuying1103@gmail.com (Y.-Y.S.); ma2589@gmail.com (C.-T.H.); 3Image Lab and Craniofacial Research Center, Chang Gung Memorial Hospital, Taoyuan 333, Taiwan; sharley@cgmh.org.tw

**Keywords:** orthognathic surgery, patient-reported outcomes, clinician-reported outcomes, observer-reported outcomes, FACE-Q, panel assessment, facial aesthetic, psychosocial

## Abstract

Outcome measures reported by patients, clinicians, and lay-observers can help to tailor treatment plans to meet patients’ needs. This study evaluated orthognathic surgery (OGS) outcomes using pre- and post-OGS patients’ (*n* = 84) FACE-Q reports, and a three-dimensional facial photograph-based panel assessment of facial appearance and psychosocial parameters, with 96 blinded layperson and orthodontic and surgical professional raters, and verified whether there were correlations between these outcome measurement tools. Post-OGS FACE-Q and panel assessment measurements showed significant (*p* < 0.001) differences from pre-OGS measurements. Pre-OGS patients’ FACE-Q scores were significantly (*p* < 0.01) lower than normal, age-, gender-, and ethnicity-matched individuals’ (*n* = 54) FACE-Q scores, with no differences in post-OGS comparisons. The FACE-Q overall facial appearance scale had a low, statistically significant (*p* < 0.001) correlation to the facial-aesthetic-based panel assessment, but no correlation to the FACE-Q lower face and lips scales. No significant correlation was observed between the FACE-Q and panel assessment psychosocial-related scales. This study demonstrates that OGS treatment positively influences the facial appearance and psychosocial-related perceptions of patients, clinicians and lay observers, but that there is only a low, or no, correlation between the FACE-Q and panel assessment tools. Future investigations may consider the inclusion of both tools as OGS treatment endpoints for the improvement of patient-centered care, and guiding the health-system-related decision-making processes of multidisciplinary teams, policymakers, and other stakeholders.

## 1. Introduction

Orthognathic surgery (OGS) has been demonstrated to correct a wide spectrum of dentofacial deformities [1,2,3]. There is an ever-growing body of OGS outcomes research, focused not only on functional occlusion, but also on the facial appearance and psychosocial domains [4,5]. 

A variety of outcome measurement tools have been adopted to assess these domains, with an emphasis on patient-reported outcomes (PROs) [1,2,3,4,5].

The recent introduction of the FACE-Q tool, a cross-culturally developed and facial-procedure-specific PRO instrument, has revolutionized the field of facial surgery outcome research by enabling the detection of meaningful and interpretable facial features and treatment-related changes and benefits [6,7,8,9]. However, only a few OGS studies have adopted the FACE-Q tool [10,11]. In this setting, the impact of different surgical interventions, including OGS procedures, on facial aesthetic and social perceptions, has been demonstrated using the panel assessment tool, a metric that is centered on clinician-reported outcome (ClinRO, professionals using medical or dental judgments) and observer-reported outcome (ObsRO, judgments from laypersons with no formal training) principles [12,13,14,15,16,17,18,19,20,21,22,23].

Interestingly, certain FACE-Q scales and panel assessment scales recently adopted in OGS studies have interconnected concepts of interest, including facial appearance (i.e., the FACE-Q facial appraisal scales and the panel assessment’s beautiful, attractive, and pleasant facial aesthetic scales) and psychosocial (i.e., the FACE-Q psychosocial scales and the panel assessment’s psychosocial perception of personality traits and emotional expressions scales) concepts [10,11,17,18,19,20,21,22,23]. So far, no OGS study has applied the PRO-based FACE-Q and ClinRO- and ObsRO-based panel assessment measurement tools in the same cohort of patients with a dentofacial deformity. Moreover, we are not aware of any investigation focused on the possible correlations between these outcome measurement tools. Understanding the multidimensional impact of OGS treatment by comprehending PRO-, ClinRO-, and ObsRO-based tools may support professionals (psychologists, dentists, orthodontists, ear, nose, and throat surgeons, plastic surgeons, head and neck surgeons, oral surgeons, and maxillofacial surgeons) working in multidisciplinary teams to provide better counseling to patients and family members, to set the expectations of preoperative patients with respect to facial appearance and psychosocial aspects, and to anticipate potential postoperative care profiles for patients with the early establishment of psychosocial support.

The primary purpose of this study was to assess the pre- versus post-OGS treatment outcomes using FACE-Q facial appearance and psychosocial reports and ClinRO- and ObsRO-based panel assessments of facial aesthetic, personality trait, and emotional expression parameters. A secondary purpose was to verify whether or not there are correlations between these outcome measurement tools.

## 2. Patients and Methods

A comparative cross-sectional study was performed, as shown in Figure 1, on patients with a dentofacial deformity (skeletal Class II and III deformities) who were managed by the same multidisciplinary team following standard pre- and post-orthognathic surgery (OGS) treatment principles [24,25,26,27,28] between 2016 and 2017. Demographic, clinical, and outcome (FACE-Q and panel assessment tool) data were collected from the Chang Gung Craniofacial Research Center’s database. Patients with an abnormal mentality that would impair the instrument’s application, patients with a normal occlusion, any syndromic diagnosis, or who had previously undergone facial surgery or a facial aesthetic procedure, and patients with an incomplete recording or postoperative follow-up (<12 months), were excluded from this study.

The study was approved by the Institutional Review Board (IRB no. 104-A253B) and conducted in compliance with the 1975 Declaration of Helsinki as amended in 1983. Patients provided written consent for the use of their images.

### 2.1. FACE-Q Tool

Taiwanese Chinese patients completed the validated Mandarin Chinese version of FACE-Q [29] during clinical appointments before or after (>12 months) an OGS procedure. Five scales were applied, namely three in the facial appraisal domain (the satisfaction with facial appearance overall, satisfaction with lower face and jawline, and satisfaction with lips scales), and two in the quality of life domain (the social function and psychological well-being scales) [6,7,8,9,10,11]:(a)Satisfaction with facial appearance overall: Measures patient satisfaction with the overall appearance of their face.(b)Satisfaction with lower face and jawline: Measures patient satisfaction with their lower face and jawline.(c)Satisfaction with lips: Measures patient satisfaction with their lips.(d)Social function: Has a series of positively-worded statements that measure social confidence.(e)Psychological well-being: Measures psychological well-being in terms of a series of positively-worded statements.

All FACE-Q scales ask patients to answer items with facial appearance in mind. The sum score for each scale was converted to an equivalent Rash score, ranging from 0 to 100, with higher values indicating a greater satisfaction with appearance or superior quality of life [6,7,8,9,10,11].

Normal Taiwanese Chinese individuals’ FACE-Q reports (no history of facial deformity, trauma, or surgery) were retrieved from the Chang Gung Craniofacial Research Center’s database, matched for age and gender, and adopted for a comparative analysis.

### 2.2. Three-Dimensional (3D)-Image-Based Panel Assessment Tool

Three-dimensional frontal and profile photographic imaging data of preoperative and postoperative (>12 months) patients were acquired using the 3dMD system (3dMD LLC, Atlanta, GA, USA) under standard conditions (a permanent installation with a fixed ambient lighting system and individuals in a fixed position with a natural head position, a relaxed facial musculature, a closed mouth, and wearing a thin elastic nylon cap to keep the hair away from the face) [30]. Using the 3dMD Vultus software (version 2.2, 3dMD Inc., Atlanta, GA, USA), a standard positioning of the three-dimensional facial images was achieved through the use of soft tissue reference planes that are meaningfully correlated to craniofacial skeleton orientation [30,31]. The system was calibrated before the image capture process.

Presentations (colored slides with frontal and profile views of the right and left sides, respectively) were delivered using PowerPoint for Mac (Microsoft Corporation, Redmond, WA, USA) on a 15-inch MacBook Pro (Apple, Inc., Cupertino, CA, USA). All preoperative and postoperative image slides were randomly distributed and rated by a panel composed of 96 raters, with no previous or current relationship to the patients using previously-published 7-point Likert scales [17,18,19,20,21,22,23]. We used three scales for facial aesthetic parameters (beautiful, attractive and pleasant), five social scales for personality trait parameters (intelligent, friendly, threatening, trustworthy and dominant), and six social scales for emotional expression parameters (angry, surprised, happy, sad, afraid and disgusted) (see Appendix A) [17,18,19,20,21,22,23]. For the ObsRO assessment, 72 laypersons (36 women, aged 18–27 years old) with no specialized professional training (i.e., no dental, medical, or psychology background) were randomly recruited based on incidental contacts from members of the general community. For the ClinRO assessment, 24 professionals (12 women) with dental or surgical training (12 orthodontics and 12 plastic surgeons) were randomly selected from the Taiwan Association of Orthodontists and the Taiwanese Society of Plastic Surgery. All raters received the same instruction and guidance before their appraisal of the 3D image set. Using one spreadsheet per slide, the rater wrote down (marking a circle corresponding to a choice from 1 to 7 on a 7-point Likert scale) his/her perceptions of the patient under appraisal with respect to the facial aesthetic, personality trait and emotional expression parameters. Raters were blinded to the purpose of the study, masked to the operative status of each image, and were not permitted to go back in the presentation. Ten percent of the images were randomly replicated for intra-rater reliability. The scores (1–7) for each scale were averaged for all pre- and post-OGS photographs, and were then adopted in the analysis.

### 2.3. Statistical Analysis

For the descriptive analysis, the mean was used for metric variables, and percentages were used for categorical variables. The data distribution was verified by the Kolmogorov–Smirnov test. The Wilcoxon signed-rank, Kruskal–Wallis, Spearman’s correlation, Cronbach’s Alpha, and intraclass correlation coefficient (ICC) tests were used for the analysis [32,33,34,35,36,37]. A Bonferroni correction was applied for multiple comparisons. The correlations among the FACE-Q scales were predicted to be moderate because these scales measure different but related features. For FACE-Q scales versus panel-assessment-related scales, the correlations were predicted to be low or non-significant, as these outcome measurement tools assess different constructs within the broader facial appearance and psychosocial domains. Spearman’s rank correlation coefficients were interpreted as high (*r* > 0.70), moderate (*r* = 0.30–0.70), and low (*r* < 0.30). Two-sided values of *p* < 0.05 were considered statistically significant. All analyses were performed using SPSS version 20.0 (Chicago, IL, USA).

## 3. Results

Eighty-four patients (22.4 ± 1.4 years of age at time of the FACE-Q report, 50% females, and 84% with a skeletal class III deformity) and 54 normal, age-, gender-, and ethnicity-matched individuals were included in this study (Table 1).

### 3.1. FACE−Q Tool

Post−OGS FACE−Q facial appraisal (satisfaction with facial appearance overall, satisfaction with lower face and jawline, and satisfaction with lips scales) and quality of life (social function and psychological well−being scales) scores were significantly (*p* < 0.05) higher than pre−OGS scores. The pre−OGS scores were significantly (*p* < 0.05) lower than the normal individuals’ scores. No significant difference was found in the comparisons between patients’ post−OGS reports and normal individuals’ reports (Table 1).

### 3.2. Panel Assessment Tool

All post−OGS facial aesthetic, personality trait, and emotional expression scores were significantly (*p* < 0.05) different from the pre−OGS scores (Table 2, Table 3 and Table 4; Figure 2 and Figure 3). No significant differences were found in the comparisons of surgeons versus orthodontics, laypersons versus surgeons, and laypersons versus orthodontics scores (Table 2, Table 3 and Table 4). The intra−and inter−rater reliabilities were good to excellent for the three groups of raters (ICC = 0.62–0.95), with low, positive correlations (*r* = 0.02–0.28; *p* < 0.05) between layperson, surgeon, and orthodontic raters (see Appendix A).

### 3.3. Correlation Evaluation

For the FACE−Q tool, significant (*p* < 0.05, moderate coefficients) correlations were observed between the facial appraisal (satisfaction with facial appearance overall, satisfaction with lower face and jawline, and satisfaction with lips scales) and quality of life (social function and psychological well−being scales) domains (Table 5).

For the panel assessment tool, the facial aesthetic scales for the beautiful and attractive parameters demonstrated significant (*p* < 0.05, low−to−moderate coefficients) correlations in all three groups of raters (Table 6; see also Appendix A). The “pleasant” parameter had significant (*p* < 0.05, low coefficients) correlations for pre−OGS clinicians’ scores and non−significant correlations for post−OGS clinicians’ scores and pre− and post−OGS observers’ scores (Table 6; see also Appendix A). No significant correlations were observed for the 11 social scales (personality trait and emotional expression parameters) (Table 7; Table 8; see also Appendix A).

The panel assessment of facial aesthetics presented significant (*p* < 0.05, low correlation coefficients) correlations with the FACE−Q satisfaction with the overall facial appearance scale, but had no significant correlation with the satisfaction with the lower face and jawline and satisfaction with lips scales (Table 6; see also Appendix A). There was no significant correlation between the panel assessment of personality traits and emotional expressions and the FACE−Q social function and psychological well−being scales (Table 7 and Table 8); see also Appendix A).

## 4. Discussion

The importance of PRO−, ClinRO−, and ObsRO−based measurement tools for OGS treatment outcomes is increasingly being recognized [1,2,3,4,5,10,11,17,18,19,20,21,22,23]. A standard outcome measurement set should be patient−centered and focus on metrics that matter most to patients, but must also be holistic and encompass important outcome domains, including complementary clinician and observer metrics [1,2,3,4,5,10,11,17,18,19,20,21,22,23]. However, to date, the PRO−based FACE−Q tool and the ClinRO− and ObsRO−based panel assessment tools have not been combined as a single outcome measurement set in the same cohort of OGS−treated patients.

In previous cross−sectional studies, OGS treatment was found to improve FACE−Q reports with respect to the facial appraisal and quality of life scales [10,11]. It has also been demonstrated that the ObsRO−based panel assessment scores significantly change after OGS treatment with respect to the perceptions of facial aesthetics and personality trait and emotional expression parameters [17,18,19,20,21,22,23]. In this study, we adopted a validated and reliable PRO−based FACE−Q tool and a high−quality, high−precision 3D facial surface image−based panel assessment. Our current results reinforce these previous findings [10,11,17,18,19,20,21,22,23], as patients’ and laypersons’ post−OGS scores were significantly different from their pre−OGS measurements. We also contribute to the literature on OGS by demonstrating the significant impact of OGS treatment on facial aesthetics, personality traits, and emotional expressions as perceived by clinicians with dental and surgical backgrounds. Additionally, using normal individuals’ FACE−Q data as a reference point, we revealed substantial modifications to before and after OGS treatment scores for all tested scales.

Previous non−OGS investigations have shown that facial personality traits can influence careers, financial success and political leadership [38,39,40,41,42]. Facial emotional expressions play a key role in guiding social judgments, including deciding whether or not to approach another person [37,38,39,40,41,42,43]. Importantly, abnormalities in these facial−aesthetics−based social judgments can result in different degrees of socially inappropriate and risky behavior [37,38]. Interestingly, a comparative analysis revealed a positive change in personality traits and emotional expressions in OGS−treated patients compared to their peers who had not undergone an OGS procedure. Additionally, our findings, along with previous results [10,11,17,18,19,20,21,22,23], suggest that the overall effect of OGS treatment has the potential to improve patients’, clinicians’ and observers’ perceptions across many aspects of facial appearance and social interaction. Further investigation is necessary to verify if FACE−Q and panel assessment tools can predict these socially relevant parameters with important real−world consequences in the OGS population.

In this investigation, we also explored the question of whether the commonly used ClinRO− and ObsRO−based panel assessment tools have any correlation to the recently developed PRO−based FACE−Q tool. In the literature, the presence or absence of associations between two different outcome measurement tools has implications for the ongoing discussions on how to interpret and apply each existing measurement tool in clinical practice and research [44,45,46,47]. 

For our study, the tests for correlations were based on predefined propositions about expected correlations, as they may attenuate the risk of bias for the described results as well as enable us to avoid alternative explanations after data collection and analysis. The potential correlations between the FACE−Q and panel assessment tools were established with respect to the facial appearance and psychosocial domains. To support the testing for correlations, for example, in the psychosocial domain, we adopted propositions from previous studies: (1) A point that patients repeatedly made during the development of the FACE−Q psychosocial well−being scale was to feel more confidence at different levels of social interactions (including in group situations or with strangers) after facial treatment [6]; and (2) the panel assessment of social perceptions was justified by the assumption that the included groups of raters are representative of the persons with whom the patients may randomly interact on a daily basis [17,18,19,20,21,22,23].

As shown in Table 5, the correlations between the FACE−Q scales had moderate correlation coefficients. This was also demonstrated by the original developers of the FACE−Q tool [6,7,8,9]. It reinforces that each specific FACE−Q scale measures particular features that matter to patients [6,7,8,9,10,11]. With regard to the remaining correlations, only the FACE−Q satisfaction with the facial appearance overall scale had significant correlations with the panel assessment of facial aesthetics; however, it only had a low correlation coefficient. Furthermore, we found no significant correlation between the facial−aesthetic−related “pleasant” parameter and some groups of raters (Table 6; see also Appendix A), suggesting that the tested tools were appraising the facial appearance domain in a different way. The FACE−Q satisfaction with the facial appearance overall scale is composed of multiple items that were carefully selected from a pool of potential items using advanced qualitative and quantitative research methods [6,7,8,9]. This instrument development process resulted in a particular FACE−Q scale that captures patients’ satisfaction with facial appearance from a global perspective, with no focus on particular anatomical regions of the face [6,7,8,9]. The panel assessment of facial aesthetics also represents an appraisal of patients’ face photographs from a global perspective. However, the unidimensional beautiful, attractive, and pleasant scales [17,18,19,20,21,22,23] are not as comprehensive as the FACE−Q satisfaction with the facial appearance overall scale [6,7,8,9].

We did not find a significant correlation between the FACE−Q satisfaction with the lower face and jawline and the satisfaction with the lips scales and the panel assessment of facial aesthetics. As these FACE−Q scales were specifically developed to capture facial−appearance−related details for each facial anatomical area [6,7,8,9], the patients provided a score for each particular scale mainly by considering specific regions of the face. In contrast, due to the characteristics of facial aesthetic scales (generic and unidimensional features) adopted for panel assessment [17,18,19,20,21,22,23], it is plausible to suppose that raters (including professionals who specialize in surgical and orthodontic fields) primarily rated the face as an overall unit, with the lower face and lips regions not necessarily being considered as targets of appraisal. Further studies should further investigate this issue with the inclusion of a panel assessment of both full−face and cropped facial images (lower face and lip regions), by using facial anatomical region−specific scales (e.g., lip attractiveness).

Our results show that the panel assessment of personality traits and emotional expressions did not have a significant correlation with the FACE−Q social and psychological well−being scales. Similar explanations regarding the dimensionality and comprehensiveness of the tested scales as those detailed above may be applied to these findings. Furthermore, as the psychosocial domain should be interpreted as being represented by an integrated biopsychosocial model of health status that accounts for the complex interplay, not only of psychological factors, but also of sociodemographic and environmental components [48], several features not directly measured in our study may have influenced patients’, clinicians’ and observers’ perceptions regarding pre− and post−OGS treatment status as well as the tested correlations. As the panel assessment was an indirect appraisal of patients’ faces, raters’ judgments may have been influenced by their prior experiences as well as other non−controlled−for factors (e.g., eyes and beard aspects), that can lead to scoring that is unconnected to OGS−treatment−related features. On the other hand, the FACE−Q reports are probably subject to a marked influence from the expectations and results (pre−OGS and post−OGS measurements, respectively) of the OGS treatment itself on the patients’ scores. 

Therefore, while the results of our panel assessment, as well as the results of previous studies [17,18,19,20,21,22,23], demonstrated that post−OGS patients’ facial images were perceived to be more friendly, happy, trustworthy, intelligent and dominant and less threatening, angry, sad, afraid and disgusted, the relationship between these parameters and the patients’ perspective about themselves in a “real−world environment” should be verified in future studies using study designs with alternate methodologies.

This study is not without limitations. As only patients matched for age, gender and type of skeletal deformity were included, a relatively small final sample was adopted in the analysis. The number of enrolled FACE−Q reports and facial photographs was, however, superior to previous studies evaluating similar outcome measurement tools [10,11,17,18,19,20,21,22,23]. Moreover, our cohort was constituted by patients with no stratification per facial appearance as judged by the authors or treating professionals, reducing the bias related to this analysis based only on the “best surgical results” [19,20]. Extrapolations from the current findings on the impact of OGS treatment on the tested parameters should be carefully made. This study was grounded on patients who were managed by senior professionals (orthodontists and surgeons) working in a multidisciplinary OGS team with specific technical strategies, such as 3D simulation, a digital occlusion setup, a surgery−first model, the single−splint, two−jaw surgery technique, and modified face bow principles [24,25,26,27,28]. The context of data collection should also be considered when interpreting these results, as facial aesthetics, personality traits and emotional expressions are perceived differently by individuals of different cultural backgrounds [49,50]. There are particular nuances for facial−appearance−related treatments and appraisals in Asians compared to Caucasians [51,52].

Similarly to previous PRO−based cross−sectional studies [10,11,53], the relationships identified using correlation coefficients should be interpreted as associations and not causal relations [33]. The present study may act as a data reference to generate hypotheses that justify further investigations. Regarding the panel assessment tool, we did not include professionals with different training backgrounds. In contrast to previous OGS studies that used the panel assessment tool [17,18,19,20,21,22,23], we have divided the raters into two groups: Clinicians and observers. Each group of raters demonstrated good to excellent intra− and inter−rater reliability, indicating that the panel assessment data were consistently collected. However, only low correlations were observed between lay observers and clinicians, indicating that the presence of a background in surgery or orthodontics may have some influence on an appraisal of facial aesthetic, personality trait and emotional expression features. Based on our findings, we suggest that some of the results from panel assessments in previous studies on these same features should be cautiously interpreted, as no explicit criteria were adopted to separate raters with and without specialized training [17,18,19,20,21,22,23]. Further research may increase the number of observers and also include OGS−treated patients and other professionals (e.g., general dentists, psychologists, ear, nose and throat surgeons, head and neck surgeons, oral surgeons, and maxillofacial surgeons) as raters to help us better understand these OGS outcomes. Other groups are also encouraged to assess their OGS cohorts to verify and expand our findings by enrolling a large sample of patients managed with a different orthodontic–surgical approach, as well as by performing further analyses, including an evaluation of the potential impact of independent variables (e.g., sociodemographic, clinical and surgical information) on FACE−Q and panel assessment tools. Further OGS outcome measurement tools, such as alternative scales for panel assessment (e.g., perceptions of symmetry, presence of lip cant, and harmony of smile), may also be tested.

We have addressed the social aspects of face perception (e.g., attractiveness and emotional expressions) as they have consistently been recognized as fundamental to human social interaction [12,13,14,15,17,18,19,20,21,22,23,38,39,40,41,42,43,54,55,56,57]. However, research on the OGS population should be expanded to additional facial−perception−associated areas of interest, such as neuropsychological mechanisms, through a multidisciplinary research collaboration with groundbreaking engineering technologies (e.g., brain scanning, direct stimulation of the brain, visual adaptation, and single−cell recording) [58,59,60].

Despite these shortcomings, the results of the present study enable us to provide practical suggestions for future OGS−outcome−based research and clinical practice. Institutions, clinicians, healthcare networks, and policymakers use results from clinical trials as the foundation for healthcare decision−making when managing individual patients or particular populations [44,45,46,47,61,62,63,64]. To design valid and meaningful clinical trials, the fundamental issue is “which form of therapeutic management presents the highest possibility of being more beneficial for the least cost and inconvenience (i.e., risk–benefit ratio) to the patient as well as to the provider?” [61]. The selection of a proper outcome measurement tool is, therefore, a key component that influences the value of outcome−based research [44,45,46,47,61,62,63,64]. As both the panel assessment tool and the FACE−Q tool were found to be capable of distinguishing patients before and after OGS treatment, consistent with previous findings [10,11,17,18,19,20,21,22,23], the lack of statistical significance for most of the tested correlations was probably not related to an inability of each tool to capture relevant factors connected with the patient (facial appearance and social factors), the disease (dentofacial deformity) and OGS treatment. In addition, considering the inherent bias in, and limitations to, each outcome measure tool, recent literature has counseled that it can be advantageous to use different tools to complement one another [44,45,46,47]. We may, therefore, advocate for the adoption of panel assessment and FACE−Q tools, either in isolation (this is acceptable if the study is constructed over a well−defined hypothesis and the restrictions of each tool are accepted) or in combination (two or more outcome measurement tools), but not as interchangeable tools. As such, capturing FACE−Q data would be a valuable addition to a panel assessment (and vice−versa) as, in fact, one outcome measurement tool may provide useful and complementary information beyond that provided by another one about the domains under consideration. For this, it is of paramount importance that each specific tool is appropriately selected using an a priori hypothesis regarding the clinical scenario or treatment outcome.

For clinical practice, the integration of ClinRO− and ObsRO−based metrics with appropriate PRO−based measures should allow for multidisciplinary teams to move toward patient−centered care. It is reasonable to better educate future OGS patients on the differences among clinicians’, observers’ and actual patients’ perspectives, as they reflect dissimilar but complementary contexts. FACE−Q data may help health−care professionals predict how future patients are likely to react to a range of concepts within the facial appearance and psychosocial domains, while a panel assessment may help these professionals understand how a patient’s facial appearance is likely to generate perceptions among the general public and clinicians. Policymakers and other stakeholders may also apply the current findings in strategic, science−driven and health−system−related decision−making processes, and to guide investment decisions for the management of patients with facial deformities and malocclusion.

## 5. Conclusions

This study demonstrates that: (1) OGS treatment positively influences patients’ facial appearance and psychosocial perceptions, as well as clinicians’ and lay observers’ perceptions of the facial aesthetics, personality traits, and emotional expressions of OGS patients; and (2) there is a low, or no, correlation between the FACE−Q and panel assessment tools.

## Figures and Tables

**Figure 1 jcm-08-00909-f001:**
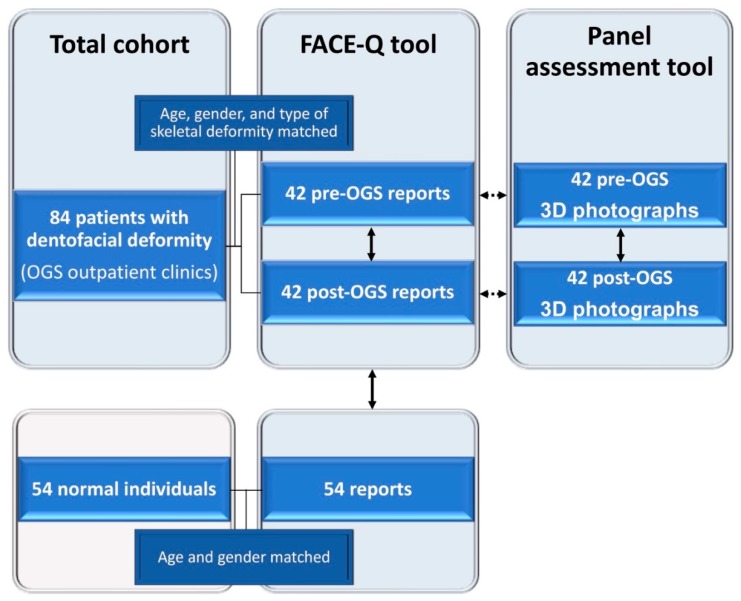
A flowchart for data collection in the comparative cross-sectional study using FACE-Q (pre- and post-orthognathic surgery (OGS) patient and normal individual reports) and panel assessment (clinicians’ and observers’ perceptions of patients’ three-dimensional (3D) facial photographs) tools. Solid line arrows and dotted line arrows indicate comparison and correlation analyses, respectively.

**Figure 2 jcm-08-00909-f002:**
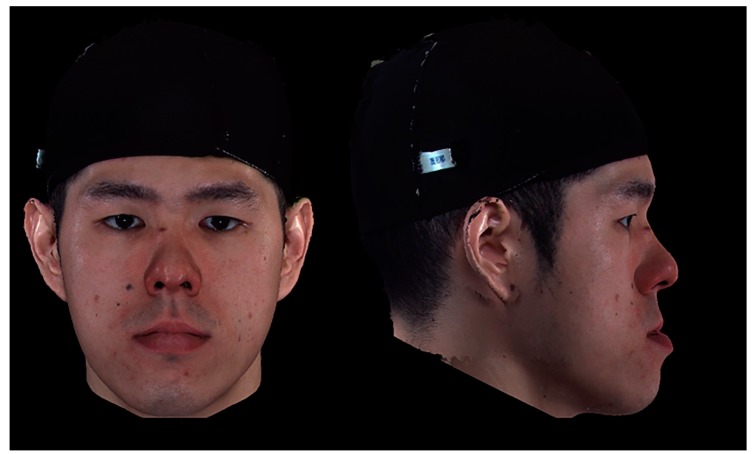
An example of a three−dimensional facial image of a male patient with a skeletal class III deformity pre−orthognathic surgery. On panel assessment, the average perceived facial personality traits were 3.6, 3.1, and 3.3 (lay observer, orthodontic professional, and surgical professional, respectively); 3.4, 3.7, and 3.2; 3.8, 4.1, and 3.9; 3.4, 3.3, and 3.8; and 3.1, 3.7, and 3.5 for the intelligent, friendly, threatening, trustworthy, and dominant parameters, respectively.

**Figure 3 jcm-08-00909-f003:**
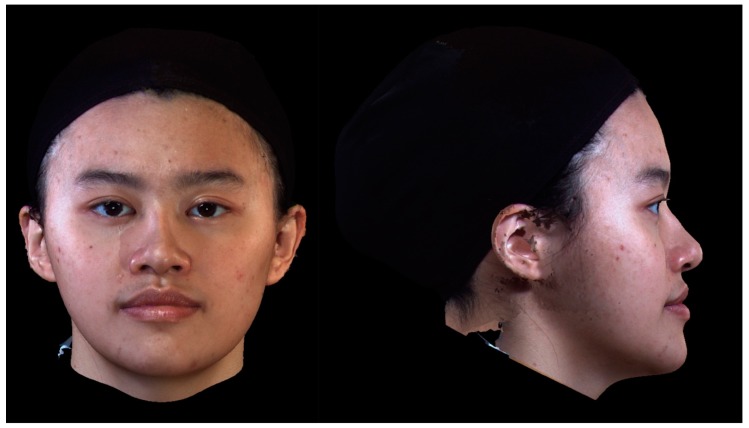
An example of a three−dimensional facial image of a female patient with a skeletal class III deformity post−orthognathic surgery. On panel assessment, the average perceived facial personality traits were: 4.7, 4.4, and 4.8 (lay observer, orthodontic professional, and surgical professional, respectively); 4.8, 5.1, and 4.7; 3.1, 3.5, and 3.4; 5.1, 4.9, and 5.3; and 5.2, 4.9, and 5.4 for the intelligent, friendly, threatening, trustworthy, and dominant parameters, respectively.

**Table 1 jcm-08-00909-t001:** Characteristics and FACE-Q reports of patients and normal individuals.

Parameters	Pre-OGS Assessment (*n* = 42)	Post-OGS Assessment (*n* = 42)	Normal Individuals (*n* = 54)	*p* *	*p* **	*p* †
**Age (y)** m ± s.d.	21.1 ± 1.4	22.3 ± 1.5	22.5 ± 1.0	>0.05	>0.05	>0.05
**Females***n* (%)	21 (50)	21 (50)	27 (50)	>0.05	>0.05	>0.05
**FACE-Q scales** m ± s.d. (95% CI)						
Facial appearance overall	45.2 ± 19.5	58.8 ± 17.1	56.9 ± 15.5	**<0.001**	**<0.001**	>0.05
	(39.2−51.3)	(53.4−64.1)	(52.6−61.1)			
Lower face and jaw line	35.6 ± 27.0	67.8 ± 18.3	65.1 ± 21.6	**<0.001**	**<0.001**	>0.05
	(27.2−44.0)	(62.1−73.5)	(59.2−71.0)			
Lips	47.7 ± 21.6	64.5 ± 22.6	66.7 ± 19.4	**<0.001**	**<0.001**	>0.05
	(41.0−54.5)	(57.5−71.5)	(61.4−72.0)			
Social function	43.7 ± 21.8	56.8 ± 23.2	55.2 ± 20.9	**<0.001**	**<0.001**	>0.05
	(36.9−50.5)	(49.5−64.0)	(49.5−60.9)			
Psychological well−being	53.8 ± 24.6	67.5 ± 23.4	69.4 ± 20.1	**<0.001**	**<0.001**	>0.05
	(46.1−61.4)	(60.2−74.9)	(63.9−74.9)			

Bold values indicate statistical significance after Bonferroni correction. y, years; m, mean; s.d., standard deviation; *n*, number of individuals; CI, confidence interval; *p*, *p*-value; OGS, orthognathic surgery; *, pre−OGS versus post−OGS comparisons; **, pre−OGS versus normal comparisons; †, post−OGS versus normal comparisons.

**Table 2 jcm-08-00909-t002:** Panel assessment of facial aesthetics.

Facial Aesthetic Scales	Pre−OGS Assessment	Post−OGS Assessment	*p* **
**Beautiful** m ± s.d. (95% CI)			
Lay observer	2.17 ± 0.89 (2.12−2.22)	3.95 ± 1.40 (3.88−4.02)	**<0.001**
Orthodontic	2.22 ± 0.90 (2.11−2.33)	4.14 ± 1.48 (3.99−4.29)	**<0.001**
Surgeon	2.36 ± 0.97 (2.24−2.48)	4.52 ± 1.50 (4.39−4.65)	**<0.001**
*p* *	>0.05	>0.05	–
**Attractive** m ± s.d. (95% CI)			
Lay observer	2.33 ± 0.98 (2.27−2.39)	4.13 ± 1.46 (4.05−4.21)	**<0.001**
Orthodontic	2.40 ± 1.09 (2.27−2.53)	4.49 ± 1.40 (4.35−4.63)	**<0.001**
Surgeon	2.54 ± 1.21 (2.40−2.68)	4.70 ± 1.55 (4.54−4.86)	**<0.001**
*p* *	>0.05	>0.05	–
**Pleasant** m ± s.d. (95% CI)			
Lay observer	2.68 ± 0.74 (2.63−2.73)	4.42 ± 2.77 (4.32−4.52)	**<0.001**
Orthodontic	2.47 ± 0.90 (2.37−2.57)	4.05 ± 1.41 (3.91−4.19)	**<0.001**
Surgeon	2.31 ± 0.87 (2.20−2.42)	4.22 ± 1.49 (4.09−4.35)	**<0.001**
*p* *	>0.05	>0.05	–

Bold values indicate statistical significance after Bonferroni correction. OGS, orthognathic surgery; m, mean; s.d., standard deviation; CI, confidence interval; *p*, *p*−value; –, not applicable; *, comparisons among the groups of raters; **, pre−OGS versus post−OGS scores comparisons.

**Table 3 jcm-08-00909-t003:** Panel assessment of personality traits.

Personality Trait Scales	Pre−OGS Assessment	Post−OGS Assessment	*p* **
**Intelligent** m ± s.d. (95% CI)			
Lay observer	3.47 ± 1.46 (3.41−3.53)	4.96 ± 1.59 (4.89−5.05)	**<0.001**
Orthodontic	3.13 ± 1.50 (3.01−3.27)	4.54 ± 1.82 (4.37−4.69)	**<0.001**
Surgeon	3.24 ± 1.26 (3.14−3.36)	4.63 ± 1.79 (4.48−4.79)	**<0.001**
*p* *	>0.05	>0.05	–
**Friendly** m ± s.d. (95% CI)			
Lay observer	3.26 ± 1.38 (3.22−3.32)	4.73 ± 1.71 (4.64−4.82)	**<0.001**
Orthodontic	3.52 ± 1.53 (3.39−3.66)	5.26 ± 1.83 (5.11−5.43)	**<0.001**
Surgeon	3.32 ± 1.54 (3.19−3.46)	4.89 ± 1.61 (4.75−5.03)	**<0.001**
*p* *	>0.05	>0.05	–
**Threatening** m ± s.d. (95% CI)			
Lay observer	3.66 ± 1.70 (3.58−3.74)	3.12 ± 1.51 (3.06−3.18)	**<0.001**
Orthodontic	3.91 ± 1.49 (3.78−4.04)	3.41 ± 1.39 (3.28−3.53)	**<0.001**
Surgeon	3.80 ± 1.28 (3.69−3.91)	3.34 ± 1.55 (3.21−3.48)	**<0.001**
*p* *	>0.05	>0.05	–
**Trustworthy** m ± s.d. (95% CI)			
Lay observer	3.22 ± 1.64 (3.15−3.29)	4.91 ± 1.99 (4.82−5.02)	**<0.001**
Orthodontic	3.40 ± 1.57 (3.27−3.54)	5.03 ± 1.95 (4.86−5.20)	**<0.001**
Surgeon	3.65 ± 1.49 (3.53−3.79)	5.19 ± 1.78 (5.04−5.35)	**<0.001**
*p* *	>0.05	>0.05	–
**Dominant** m ± s.d. (95% CI)			
Lay observer	3.29 ± 1.83 (3.22−3.36)	4.95 ± 1.85 (4.87−5.03)	**<0.001**
Orthodontic	3.53 ± 1.51 (3.40−3.66)	5.15 ± 1.89 (4.98−5.31)	**<0.001**
Surgeon	3.70 ± 1.47 (3.57−3.83)	5.37 ± 1.75 (5.21−5.52)	**<0.001**
*p* *	>0.05	>0.05	–

Bold values indicate statistical significance after Bonferroni correction. OGS, orthognathic surgery; m, mean; s.d., standard deviation; CI, confidence interval; *p*, *p*−value; –, not applicable; *, comparisons among the groups of raters; **, pre−OGS versus post−OGS scores comparisons.

**Table 4 jcm-08-00909-t004:** Panel assessment of emotional expressions.

Emotional Expression Scales	Pre−OGS Assessment	Post−OGS Assessment	*p* **
**Angry** m ± s.d. (95% CI)			
Lay observer	2.73 ± 1.33 (2.65−2.81)	2.30 ± 1.36 (2.26−2.38)	**<0.001**
Orthodontic	2.77 ± 1.40 (2.63−2.91)	2.41 ± 1.25 (2.29−2.53)	**<0.001**
Surgeon	2.64 ± 1.90 (2.51−2.79)	2.23 ± 1.18 (2.13−2.33)	**<0.001**
*p* *	>0.05	>0.05	–
**Surprised** m ± s.d. (95% CI)			
Lay observer	2.61 ± 1.49 (2.56−2.68)	2.99 ± 1.77 (2.91−3.07)	**<0.001**
Orthodontic	2.59 ± 1.35 (2.45−2.73)	2.91 ± 1.19 (2.80−3.02)	**<0.001**
Surgeon	2.45 ± 1.43 (2.34−2.56)	2.83 ± 1.48 (2.72−2.96)	**<0.001**
*p* *	>0.05	>0.05	–
**Happy** m ± s.d. (95% CI)			
Lay observer	3.11 ± 1.25 (3.08−3.16)	3.81 ± 2.39 (3.72−3.90)	**<0.001**
Orthodontic	3.17 ± 1.08 (3.05−3.29)	3.75 ± 1.52 (3.59−3.91)	**<0.001**
Surgeon	3.39 ± 1.55 (3.26−3.52)	3.93 ± 1.94 (3.77−4.09)	**<0.001**
*p* *	>0.05	>0.05	–
**Sad** m ± s.d. (95% CI)			
Lay observer	3.61 ± 1.24 (3.54−3.68)	3.25 ± 1.26 (3.17−3.33)	**<0.001**
Orthodontic	3.83 ± 1.45 (3.71−3.95)	3.32 ± 1.69 (3.18−3.48)	**<0.001**
Surgeon	3.77 ± 1.36 (3.66−3.90)	3.40 ± 1.58 (3.26−3.54)	**<0.001**
*p* *	>0.05	>0.05	–
**Afraid** m ± s.d. (95% CI)			
Lay observer	2.74 ± 1.27 (2.67−2.82)	2.30 ± 1.16 (2.22−2.36)	**<0.001**
Orthodontic	2.98 ± 1.34 (2.86−3.10)	2.59 ± 1.42 (2.46−2.71)	**<0.001**
Surgeon	2.86 ± 1.23 (2.74−2.98)	2.37 ± 1.30 (2.25−2.51)	**<0.001**
*p* *	>0.05	>0.05	–
**Disgusted** m ± s.d. (95% CI)			
Lay observer	3.69 ± 1.24 (3.61−3.76)	3.32±1.44 (3.24−3.41)	**<0.001**
Orthodontic	3.77 ± 1.21 (3.67−3.88)	3.38±1.40 (3.25−3.50)	**<0.001**
Surgeon	3.45 ± 1.65 (3.32−3.58)	3.19±1.59 (3.08−3.30)	**<0.001**
*p**	>0.05	>0.05	–

Bold values indicate statistical significance after Bonferroni correction. OGS, orthognathic surgery; m, mean; s.d., standard deviation; CI, confidence interval; *p*, *p*−value; –, not applicable; *, comparisons among the groups of raters; **, pre−OGS versus post−OGS scores comparisons.

**Table 5 jcm-08-00909-t005:** Correlations for Pre− and Post−Orthognathic Surgery Assessment of FACE−Q Scales.

FACE−Q Scales	Appearance Overall	Lower Face and Jaw	Lips	Social Function
	*r* (*p*)	*r* (*p*)	*r* (*p*)	*r* (*p*)
**Lower face and jaw**				
Pre−OGS	0.69 **(<0.001)**	–	–	–
Post−OGS	0.63 **(<0.001)**	–	–	–
**Lips**				
Pre−OGS	0.56 **(<0.001)**	0.68 **(<0.001)**	–	–
Post−OGS	0.61 **(<0.001)**	0.55 **(<0.001)**	–	–
**Social function**				
Pre−OGS	0.66 **(<0.001)**	0.55 **(<0.001)**	0.52 **(<0.001)**	–
Post−OGS	0.51 **(<0.001)**	0.60 **(<0.001)**	0.59 **(<0.001)**	–
**Psychological well−being**				
Pre−OGS	0.64 **(<0.001)**	0.53 **(<0.001)**	0.54 **(<0.001)**	0.69 **(<0.001)**
Post−OGS	0.57 **(<0.001)**	0.62 **(<0.001)**	0.51 **(<0.001)**	0.65 **(<0.001)**

Bold values indicate statistical significance after Bonferroni correction. OGS, orthognathic surgery; *r*, correlation coefficient; *p*, *p*−value; –, not applicable.

**Table 6 jcm-08-00909-t006:** Correlations for Pre− and Post−Orthognathic Surgery Assessment Scores of FACE−Q Facial Appraisal Scales and Surgeon−Based Panel Assessment of Facial Aesthetic Scales.

Facial Aesthetic Scales	FACE−Q Tool			Panel Assessment Tool	
	Appearance Overall	Lower Face and Jaw	Lips	Beautiful	Attractive
	*r* (*p*)	*r* (*p*)	*r* (*p*)	*r* (*p*)	*r* (*p*)
**Beautiful**					
Pre−OGS	0.26 **(<0.001)**	0.24 (>0.05)	0.13 (<0.05)	–	–
Post−OGS	0.21 **(<0.001)**	0.18 (>0.05)	0.36 (<0.05)	–	–
**Attractive**					
Pre−OGS	0.29 **(<0.001)**	0.27 (>0.05)	0.04 (<0.05)	0.49 **(<0.001)**	–
Post−OGS	0.24 **(<0.001)**	0.09 (>0.05)	0.13 (<0.05)	0.38 **(<0.001)**	–
**Pleasant**					
Pre−OGS	0.02 **(<0.001)**	0.19 (>0.05)	0.09 (>0.05)	0.23 **(<0.001)**	0.15 **(<0.001)**
Post−OGS	0.17 **(>0.05)**	0.22 (>0.05)	0.28 (>0.05)	0.04 **(<0.001)**	0.07 **(<0.001)**

Bold values indicate statistical significance after Bonferroni correction. OGS, orthognathic surgery; *r*, correlation coefficient; *p*, *p*−value; –, not applicable.

**Table 7 jcm-08-00909-t007:** Correlations for Pre− and Post−Orthognathic Surgery Assessment Scores of FACE−Q Quality of Life Scales and Orthodontist−Based Panel Assessment of Personality Trait Scales.

Personality Trait Scales	FACE−Q Tool		Panel Assessment Tool			
	Social	Psychological	Intelligent	Friendly	Threat	Trust
	*r* (*p*)	*r* (*p*)	*r* (*p*)	*r* (*p*)	*r* (*p*)	*r* (*p*)
**Intelligent**						
Pre−OGS	0.38 (>0.05)	0.14 (>0.05)	–	–	–	–
Post−OGS	0.12 (>0.05)	0.17 (>0.05)	–	–	–	–
**Friendly**						
Pre−OGS	0.15 (>0.05)	0.33 (>0.05)	0.09 (>0.05)	–	–	–
Post−OGS	0.28 (>0.05)	0.11 (>0.05)	0.03 (>0.05)	–	–	–
**Threat**						
Pre−OGS	0.22 (>0.05)	0.13 (>0.05)	0.25 (>0.05)	0.06 (>0.05)	–	–
Post−OGS	0.26 (>0.05)	0.27 (>0.05)	0.07 (>0.05)	0.18 (>0.05)	–	–
**Trust**						
Pre−OGS	0.07 (>0.05)	0.16 (>0.05)	0.19 (>0.05)	0.22 (>0.05)	0.16 (>0.05)	–
Post−OGS	0.12 (>0.05)	0.21 (>0.05)	0.07 (>0.05)	0.13 (>0.05)	0.05 (>0.05)	–
**Dominant**						
Pre−OGS	0.17 (>0.05)	0.12 (>0.05)	0.24 (>0.05)	0.15 (>0.05)	0.10 (>0.05)	0.24 (>0.05)
Post−OGS	0.22 (>0.05)	0.34 (>0.05)	0.13 (>0.05)	0.21 (>0.05)	0.08 (>0.05)	0.19 (>0.05)

OGS, orthognathic surgery; Threat, threatening; Trust, trustworthy; *r*, correlation coefficient; *p*, *p*−value; –, not applicable.

**Table 8 jcm-08-00909-t008:** Correlations for Pre− and Post−Orthognathic Surgery Assessment Scores of FACE−Q Quality of Life Scales and Lay−Observer−Based Panel Assessment of Emotional Expression Scales.

Emotional Expression Scales	FACE−Q Tool		Panel Assessment Tool				
	Social	Psychological	Angry	Surprised	Happy	Sad	Afraid
	*r* (*p*)	*r* (*p*)	*r* (*p*)	*r* (*p*)	*r* (*p*)	*r* (*p*)	*r* (*p*)
**Angry**							
Pre−OGS	0.08 (>0.05)	0.11 (>0.05)	–	–	–	–	–
Post−OGS	0.17 (>0.05)	0.22 (>0.05)	–	–	–	–	–
Surprised							
Pre−OGS	0.14 (>0.05)	0.01 (>0.05)	0.18 (>0.05)	–	–	–	–
Post−OGS	0.20 (>0.05)	0.14 (>0.05)	0.09 (>0.05)	–	–	–	–
**Happy**							
Pre−OGS	0.27 (>0.05)	0.03 (>0.05)	0.23 (>0.05)	0.06 (>0.05)	–	–	–
Post−OGS	0.24 (>0.05)	0.26 (>0.05)	0.07 (>0.05)	0.19 (>0.05)	–	–	–
**Sad**							
Pre−OGS	0.21 (>0.05)	0.25 (>0.05)	0.04 (>0.05)	0.09 (>0.05)	0.13 (>0.05)	–	
Post−OGS	0.14 (>0.05)	0.08 (>0.05)	0.27 (>0.05)	0.13 (>0.05)	0.05 (>0.05)	–	–
**Afraid**							
Pre−OGS	0.06 (>0.05)	0.18 (>0.05)	0.22 (>0.05)	0.04 (>0.05)	0.17 (>0.05)	0.20 (>0.05)	–
Post−OGS	0.17 (>0.05)	0.11 (>0.05)	0.14 (>0.05)	0.07 (>0.05)	0.25 (>0.05)	0.03 (>0.05)	–
Disgusted							
Pre−OGS	0.13 (>0.05)	0.17 (>0.05)	0.04 (>0.05)	0.22 (>0.05)	0.09 (>0.05)	0.13 (>0.05)	0.03 (>0.05)
Post−OGS	0.07 (>0.05)	0.01 (>0.05)	0.23 (>0.05)	0.38 (>0.05)	0.17 (>0.05)	0.22 (>0.05)	0.10 (>0.05)

OGS, orthognathic surgery; *r*, correlation coefficient; *p*, *p*−value; –, not applicable.

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
