# Peer review of "Facial Appearance and Psychosocial Features in Orthognathic Surgery: A FACE-Q- and 3D Facial Image-Based Comparative Study of Patient-, Clinician-, and Lay-Observer-Reported Outcomes"

_jcm, 2019, doi:10.3390/jcm8060909_

Round 1
Reviewer 1 Report
The manuscript describes outcome measures for orthognatic surgery comparing FACE-Q and other panel assessments. The manuscript is well-written and organized. Just one major and a few minor comments:
Major comments:
All p values were presented simply as whether they met some test threshold e.g. "p>0.05" or "p<0.01" which aren't particularly informative and don't allow the reader to see the calculated statistic. The actual values calculated should be presented. E.g. p = 0.032, p = 0.003 (smaller values than 0.001 probably can be represented as p<0.001). I'm uncertain of JCM's exact policies, but similar policies are published elsewhere (https://support.jmir.org/hc/en-us/articles/360000002012-How-should-P-values-be-reported-) You may also wish to print the significant values in tables in either bold or italics.
Also the selection of p<0.05 as a test for significance should be carefully reviewed by a statistician. Multiple comparisons are being made in this manuscript and a correction like the Bonferroni correction (https://en.wikipedia.org/wiki/Bonferroni_correction) may apply. For example, assuming the Bonferroni conditions hold, Table 5 is cross-comparing 5 parameters and the threshold for significant might be 0.05/5 = 0.01 (Table 5 appears to meet this threshold) Similarly Table 6 is cross comparing 6 parameters which would be p<0.0083. I highly recommend reviewing this analysis with a statistician to determine of whether this kind of correction should apply. This may or may not affect your conclusions.
Minor comments
Line 20: I think this would look a little cleaner than the compound adjective list provided: "Outcome measures reported by patients, clinicians, and lay-observers can help ..."
Line 120: "...previously published 7-point Likert scales" Reference? I think these might be refs 17-23] which appears several lines below. I would suggest placing the references at the end of the sentence on line 120 where the studies are first mentioned and adjust the text accordingly: "...Likert scales [17-23]. These include three scales for ..."
Author Response
June 21, 2019
Prof. Dr. Emmanuel Andrès
Editor-in-Chief, Journal of Clinical Medicine
Dear Prof. Dr. Andrès,
We would like to thank you and your reviewers for considering our manuscript entitled “Facial Appearance and Psychosocial Features in Orthognathic Surgery: A FACE-Q- and 3D Facial Image-Based Comparative Study of Patient-, Clinician-, and Lay-Observer-Reported Outcomes”(jcm-528857), for publication in Otolaryngology Section of Journal of Clinical Medicine. We appreciate the thorough and thoughtful review and for all the insightful comments and suggestions, as the comments have significantly improved our manuscript. We have addressed, on a point-by-point basis, all of the comments made by the reviewers. All changes have been included (marked in red) in our revised submission.
The article was reviewed by a professional English editing company (MDPI, English-10120).
Reviewer 1#:
1. The manuscript describes outcome measures for orthognathic surgery comparing FACE-Q and other panel assessments. The manuscript is well-written and organized. Just one major and a few minor comments.
Answer: We greatly appreciate the feedback from the reviewer as the comments have significantly improved our manuscript.
2. All p values were presented simply as whether they met some test threshold e.g. "p>0.05" or "p<0.01" which aren't particularly informative and don't allow the reader to see the calculated statistic. The actual values calculated should be presented. E.g. p = 0.032, p = 0.003 (smaller values than 0.001 probably can be represented as p<0.001). I'm uncertain of JCM's exact policies, but similar policies are published elsewhere (https://support.jmir.org/hc/en-us/articles/360000002012-How-should-P-values-be-reported-) You may also wish to print the significant values in tables in either bold or italics.
Answer: All actual p-values calculated were presented in Tables for significant values, as requested.
3. Also the selection of p<0.05 as a test for significance should be carefully reviewed by a statistician. Multiple comparisons are being made in this manuscript and a correction like the Bonferroni correction (https://en.wikipedia.org/wiki/Bonferroni_correction) may apply. For example, assuming the Bonferroni conditions hold, Table 5 is cross-comparing 5 parameters and the threshold for significant might be 0.05/5 = 0.01 (Table 5 appears to meet this threshold) Similarly Table 6 is cross comparing 6 parameters which would be p<0.0083. I highly recommend reviewing this analysis with a statistician to determine of whether this kind of correction should apply. This may or may not affect your conclusions.
Answer:It was reviewed as requested. Bonferroni corrections were adopted for multiple comparisons. It did not change the interpretations of results or conclusions. It was addressed, as requested:
-Methods section: Two-sided values of p < 0.05 were considered statistically significant.
-Tables (footnotes): Adjustment for multiple comparisons (Bonferroni corrections).
4. Line 20: I think this would look a little cleaner than the compound adjective list provided: "Outcome measures reported by patients, clinicians, and lay-observers can help ..."
Answer: This sentence was reformulated as requested:
- Outcome measures reported by patients, clinicians, and lay-observers can help to tailor treatment plans to meet patients’ needs.
5. Line 120: "...previously published 7-point Likert scales" Reference? I think these might be refs 17-23] which appears several lines below. I would suggest placing the references at the end of the sentence on line 120 where the studies are first mentioned and adjust the text accordingly: "...Likert scales [17-23]. These include three scales for ..."
Answer: The references were introduced at the end of the sentence as requested.
- …patients using previously published 7-point Likert scales [17–23].
We can provide additional modifications and explanations at the request of the Reviewers and/or Editorial Board. Thank you very much for your consideration of our manuscript.
The authors

Reviewer 2 Report
Aesthetic evaluation after orthognathic surgery may be subjective, and it seems to be meaningful as a paper to compare the objective evaluation related to this.
It is recommended that you reduce the title because the title is too long for your readers to understand.
Please reinforce the explanation about FACE-Q scores.
It seems that the IRB approval number is missing for the study. Please add IRB approval number.
Author Response
June 21, 2019
Prof. Dr. Emmanuel Andrès
Editor-in-Chief, Journal of Clinical Medicine
Dear Prof. Dr. Andrès,
We would like to thank you and your reviewers for considering our manuscript entitled “Facial Appearance and Psychosocial Features in Orthognathic Surgery: A FACE-Q- and 3D Facial Image-Based Comparative Study of Patient-, Clinician-, and Lay-Observer-Reported Outcomes”(jcm-528857), for publication in Otolaryngology Section of Journal of Clinical Medicine. We appreciate the thorough and thoughtful review and for all the insightful comments and suggestions, as the comments have significantly improved our manuscript. We have addressed, on a point-by-point basis, all of the comments made by the reviewers. All changes have been included (marked in red) in our revised submission.
The article was reviewed by a professional English editing company (MDPI, English-10120).
Reviewer 2#:
1. Aesthetic evaluation after orthognathic surgery may be subjective, and it seems to be meaningful as a paper to compare the objective evaluation related to this.
Answer: We greatly appreciate the feedback from the reviewer as the comments have significantly improved our manuscript.
2. It is recommended that you reduce the title because the title is too long for your readers to understand.
Answer: The title was reduced as requested:
Facial Appearance and Psychosocial Features in Orthognathic Surgery: A FACE-Q- and 3D Facial Image-Based Comparative Study of Patient-, Clinician-, and Lay-Observer-Reported Outcomes
3. Please reinforce the explanation about FACE-Q scores.
Answer: It was addressed in Methods section, as requested:
(a) Satisfaction with facial appearance overall: measures patient satisfaction with the overall appearance of their face.
(b) Satisfaction with lower face and jawline: measures patient satisfaction with their lower face and jawline.
(c) Satisfaction with lips: measures patient satisfaction with their lips.
(d) Social function: has a series of positively worded statements that measure social confidence.
(e) Psychological well-being: measures psychological well-being in terms of a series of positively worded statements.
All FACE-Q scales ask patients to answer items with facial appearance in mind. The sum score for each scale was converted to an equivalent Rash score, ranging from 0 to 100, with higher values indicating a greater satisfaction with appearance or superior quality of life
4. It seems that the IRB approval number is missing for the study. Please add IRB approval number.
Answer: It was addressed in Methods section, as requested:
- The study was approved by the Institutional Review Board (IRB no. 104-A253B)…
We can provide additional modifications and explanations at the request of the Reviewers and/or Editorial Board. Thank you very much for your consideration of our manuscript.
The authors
